# Coronavirus-Related Health Literacy: A Cross-Sectional Study in Adults during the COVID-19 Infodemic in Germany

**DOI:** 10.3390/ijerph17155503

**Published:** 2020-07-30

**Authors:** Orkan Okan, Torsten Michael Bollweg, Eva-Maria Berens, Klaus Hurrelmann, Ullrich Bauer, Doris Schaeffer

**Affiliations:** 1Interdisciplinary Centre for Health Literacy Research, Faculty of Educational Science, Bielefeld University, 33615 Bielefeld, Germany; 2Centre for Prevention and Intervention in Childhood and Adolescence, Faculty of Educational Science, Bielefeld University, 33615 Bielefeld, Germany; torsten.bollweg@uni-bielefeld.de; 3School of Public Health, Interdisciplinary Centre for Health Literacy Research, Bielefeld University, 33615 Bielefeld, Germany; eva-maria.berens@uni-bielefeld.de; 4Department of Public Health and Education, Hertie School of Governance, 10117 Berlin, Germany; hurrelmann@hertie-school.org; 5Interdisciplinary Centre for Health Literacy Research, Centre for Prevention and Intervention in Childhood and Adolescence, Faculty of Educational Science, Bielefeld University, 33615 Bielefeld, Germany; ullrich.bauer@uni-bielefeld.de; 6Interdisciplinary Centre for Health Literacy Research, School of Public Health, Bielefeld University, 33615 Bielefeld, Germany; doris.schaeffer@uni-bielefeld.de

**Keywords:** health literacy, infodemic, survey, coronavirus 2, SARS-CoV-2, COVID-19, Germany, adult population, HLS-EU-Q

## Abstract

There is an “infodemic” associated with the COVID-19 pandemic—an overabundance of valid and invalid information. Health literacy is the ability to access, understand, appraise, and apply health information, making it crucial for navigating coronavirus and COVID-19 information environments. A cross-sectional representative study of participants ≥ 16 years in Germany was conducted using an online survey. A coronavirus-related health literacy measure was developed (HLS-COVID-Q22). Internal consistency was very high (α = 0.940; ρ = 0.891) and construct validity suggests a sufficient model fit, making HLS-COVID-Q22 a feasible tool for assessing coronavirus-related health literacy in population surveys. While 49.9% of our sample had sufficient levels of coronavirus-related health literacy, 50.1% had “problematic” (15.2%) or “inadequate” (34.9%) levels. Although the overall level of health literacy is high, a vast number of participants report difficulties dealing with coronavirus and COVID-19 information. The participants felt well informed about coronavirus, but 47.8% reported having difficulties judging whether they could trust media information on COVID-19. Confusion about coronavirus information was significantly higher among those who had lower health literacy. This calls for targeted public information campaigns and promotion of population-based health literacy for better navigation of information environments during the infodemic, identification of disinformation, and decision-making based on reliable and trustworthy information.

## 1. Introduction

The severe acute respiratory syndrome coronavirus 2 (SARS-CoV-2) is a novel virus in the coronavirus family, causing the coronavirus disease (COVID-19). It was first reported in December 2019 and has since evolved into a global pandemic that created a social and economic crisis as well as a humanitarian catastrophe [1,2,3,4]. While the human and economic costs certainly are the most significant consequences of the COVID-19 pandemic, they are tackled by challenges related to information overload and an ongoing “infodemic” [5,6].

Infodemic is short for “information epidemic”, a phenomenon that portrays the rapid spread and amplification of vast amounts of valid and invalid information on the internet or through other communication technologies [5,7]. Although the term was first coined during the SARS outbreak of 2003 [7], it has not been used much in the scientific literature before 2020 [8]. Since the beginning of the COVID-19 pandemic, both the production and consumption of information have increased rapidly and significantly [9,10]. The World Health Organization [6], amongst others [11], has stressed that the infodemic is a serious threat to public health, public action, social cohesion, and the political landscape as a whole. Baines and Elliott conclude that “(i) the infodemic is unprecedented in its size and velocity; (ii) unexpected forms of false information are emerging daily; and (iii) no global consensus exists on how best to classify the types of false messages being encountered” [12]. On the individual level, the infodemic creates confusion among recipients of information, specifically in relation to the identification of reliable information [13,14,15] as well as disinformation, misinformation, and malinformation [12]. In the current pandemic, negative information bias (causing “catastrophic thinking”) and positive information bias (causing “unrealistic optimism”) are among the many consequences and risks posed by the infodemic [16]. Altogether, this constitutes a global scientific challenge [13].

Media literacy and information literacy are critical competencies in the context of infodemics [13]. Since the current infodemic is a health-related infodemic, information literacy must be focused on health—this is known as health literacy [17]. Although health information about COVID-19 dominates most communication channels [18], health literacy has thus far remained both an underestimated concept and a neglected field of research in the context of the COVID-19 infodemic [19]. Researchers from various disciplines have urged the need to consider the health literacy of citizens, policymakers, governments, and information producers and providers [9,16,18,20,21,22].

Health literacy can facilitate distinguishing between reliable information on COVID-19 and dis- and misinformation on the topic, it helps navigating sources of health information and health services, and health literacy empowers people to make informed health decisions and to practice healthy and protective behaviors in the time of the coronavirus and COVID-19 pandemic [9,19,20]. In general, health literacy is defined as the motivation, knowledge, and competence used to access, understand, appraise, and apply health information and make health-related decisions [23]. However, health literacy is a relational concept and the system-level health literacy, e.g., the health literacy of information suppliers and service providers is as important as the health literacy of individuals [24,25,26]. Past research has shown that low health literacy is associated with various adverse health outcomes [27,28] and higher health care costs [29,30,31]. In addition, several studies have demonstrated the existence of a social gradient in low health literacy with financial deprivation, education, and social status being the strongest predictors [28,32,33]. Population-based studies on health literacy have shown that great proportions of European [28], North American [34,35], and Asian [36] populations have difficulties dealing with health-related information. The European Health Literacy Survey (HLS-EU), which was conducted in eight countries, found that almost 50% of all adults have “problematic” or “inadequate” health literacy, meaning that it is potentially difficult for them to access, understand, appraise, and apply information to promote or protect their health [28]. In Germany, this proportion is even higher, with 54.3% of all participants [37]. A recent study on fear of COVID-19 in medical students has shown that higher levels of health literacy may lower the level of fear [38]. In another study, higher health literacy levels have shown protective effects against COVID-19 related depression [39]. Understanding of public health recommendations, applying protective measures against infection with coronavirus, and navigating COVID-19-related health information environments are currently of elevated importance [18,19,20], which is why there is a need to explore coronavirus-related health literacy. A recent study in Germany is focussing on university students’ digital health literacy in relation to coronavirus and COVID-19 information [40]. This study is part of a global health literacy research network, which measures the digital health literacy, health information behavior, fear of COVID-19, and wellbeing of students during university lockdowns in 45 countries [41]. Together, these studies will contribute to creating evidence on health literacy in the time of COVID-19.

The aim of this study was to (1) test the feasibility of a newly developed assessment tool for coronavirus-related health literacy, (2) assess coronavirus-related health literacy of German adults, (3) examine differences in health literacy in relation to sociodemographic variables, and (4) explore how participants feel with regard to the overabundance of health information during the COVID-19 infodemic.

## 2. Methods

### 2.1. Study Design and Target Group

A cross-sectional online survey was conducted from 31 March to 7 April 2020 among adult internet users (≥16 years) in Germany. The sample is representative of German-speaking internet users regarding age, sex, education, and state of residence. The survey was conducted by the Interdisciplinary Centre for Health Literacy Research at Bielefeld University, Germany, in collaboration with Allensbach Institute, Allensbach, Germany. Participants were recruited by the Allensbach Institute from an online panel of 180.000 individuals, in which all participants gave their consent. Using a stratified random sampling strategy, invitations were sent by email to *n* = 4329 prospective participants, of which *n* = 1461 responded and *n* = 308 were excluded due to recruitment quota was reached. Finally, *n* = 1153 participants received a link to complete the self-administered questionnaire, which was implemented as an online survey due to COVID-19 restrictions. Participants received an incentive upon completion.

### 2.2. Measures

#### 2.2.1. Sociodemographic Indicators

Participants were asked about their sex, age, education, net household income, region of residence, and whether they had a child aged 18 or younger.

#### 2.2.2. Feeling Informed or Confused about the Coronavirus

Two general items were used to address individual perceptions of information on the coronavirus and COVID-19. First, the question “Overall, how well do you feel informed about the coronavirus or the coronavirus-epidemic?” was answered on a 4-point Likert scale: 1—“not well at all”, 2—“not so well”, 3—“well”, and 4—“very well”. The second question was: “Some people feel confused by all the information about the coronavirus or the coronavirus epidemic, and they don’t know what information to believe anymore. What about you: do you feel confused by all the information?” Responses were given on a 3-point scale: 1—“yes, very confused”, 2—“yes, somewhat confused”, and 3—“no, barely confused/not confused at all”.

#### 2.2.3. Health Literacy

Coronavirus-related health literacy was assessed based on the European Health Literacy Survey Questionnaire (HLS-EU-Q). The HLS-EU-Q is a self-report instrument for measuring a comprehensive model of health literacy [42]. It assesses participants’ perceived difficulty or ease when accessing, understanding, appraising, and applying health information [43]. Originally, the HLS-EU-Q consisted of 47 items (HLS-EU-Q47), but over the years, short forms have been developed. The 16-item version, the HLS-EU-Q16 [44], has shown high internal consistency (Cronbach’s α = 0.90) in a German representative sample [45]. The HLS-EU-Q16 comprises sixteen questions starting with “How easy or difficult is it for you to?” followed by specific questions, e.g., “…find information on treatments of illnesses that concern you?” It is answered on 4-point scale ranging from 1 (very difficult) to 4 (very easy). Usually, the questionnaire is administered as a (computer-assisted) personal interview, but telephone, paper and pencil, and online versions have been used as well [42,43]. In order to explore how German adults deal with information regarding coronavirus and COVID-19 during the pandemic, the German HLS-EU-Q16 [44,45] was adapted. After the first draft was commented and approved by the authors, it was sent to the Allensbach Institute for critical review by experts in questionnaire development. The comments were discussed and consented by the study group and the Allensbach Institute. The final questionnaire, HLS-COVID-Q22 [46], comprises 22 items organized in four subscales: accessing (six items), understanding (six items), appraising (five items), and applying (five items) health-related information in the context of the coronavirus pandemic.

HLS-COVID-Q22 mean scores are based on the responses ranging from 1 to 4, meaning that no transformation of scores was conducted, as suggested by Sørensen and colleagues [28,42]. In this way, mean scores can easily be interpreted based on the response range. The scoring system suggested in the original HLS-EU study [37,42,43,44,45] was adapted: Sørensen and colleagues [42] propose three cut-off values resulting in four levels of health literacy (Table 1). However, Röthlin et al. (2013) have found limited discrimination of the HLS-EU-Q16 between participants in the higher-end range of scores [44]. Thus, they have suggested merging the highest health literacy levels, discarding the highest cut-off value and the level “excellent health literacy”. In this study, we use the following cut-off values and levels:Mean score of ≤2.5: “inadequate health literacy”Mean score of >2.5–<3: “problematic health literacy”Mean score of ≥3: “sufficient health literacy”

### 2.3. Data Analyses

By implementing iterative proportional fitting (IPF), the dataset was weighted for sex, education, and age based on the German Mikrozensus 2018 [47]. Confirmatory factor analysis (CFA) was performed using SPSS AMOS 22 (IBM, SPSS Inc., Chicago, IL, USA) [48]. Commonly reported cut-off values were used for the model fit [49], which was evaluated using the following indices: relative χ^2^(χ^2^/df ≤ 3), the normed-fit index (NFI ≥ 0.90), the incremental fit index (IFI ≥ 0.90), the comparative fit index (CFI ≥ 0.90), the Tucker-Lewis index (TLI ≥ 0.90), the root-mean-square error of approximation (RMSEA ≤ 0.05–0.08), and the standardized root mean square residual (SRMR ≤ 0.1). Factor loadings were interpreted based on the following cut-offs, as suggested by Tabachnick and Fidell: 0.32 (poor), 0.45 (fair), 0.55 (good), 0.63 (very good), and 0.71 (excellent) [50]. Internal consistency was examined by calculating Cronbach’s α and the Spearman-Brown split-half reliability coefficient using SPSS Statistics 22 (IBM, SPSS Inc., Chicago, IL, USA) [51]. For the latter indicators, values ≥ 0.7 indicate satisfactory reliability [52]. Mean scores and relative frequencies are reported for health literacy levels and HLS-COVID-Q22 items. Bivariate correlations are reported for the relationship between HLS-COVID-Q22 subscales. Also, *p*-values for nonparametric correlations between the items on feeling informed or confused, and sociodemographic indicators are reported. Bivariate analyses were employed to calculate correlations: the Spearman-Rho correlation coefficient was used for ordinal variables; the Pearson correlation coefficient was used for metric variables. Differences in mean scores related to sociodemographic indicators were checked for significance with the Mann-Whitney-U test, as the assumptions of normality and homoscedasticity had been violated for HLS-COVID-Q22 mean scores in most groups.

### 2.4. Ethical Considerations

This study was approved by the Bielefeld University Ethics Board (Reference No 2020-060_S). All personal data obtained was anonymized by Allensbach, and the Bielefeld University study group received a fully anonymized dataset. Data protection procedures were covered in adherence to the European Union General Data Protection Regulation (EU) (GDPR). This study was carried out by the Interdisciplinary Centre for Health Literacy Research and within the Health Literacy in Childhood and Adolescence (HLCA) Consortium (www.hlca-consortium.com), funded by the German Federal Ministry of Education and Research (grant number: 01EL1824A).

## 3. Results

### 3.1. Sample Characteristics

A total of *n* = 1153 participants received a link to complete the self-administered questionnaire, of which *n* = 99 participants dropped out before completing the whole survey, and *n* = 17 were excluded because of concerns about the reliability of the data. Finally, a total of *n* = 1037 adults aged ≥16 years in all 16 German federal states were interviewed. The mean age was 45.6 years (SD = 15.8). 31.5% of the participants reported a household income under 1750 €, and between 1750–2999 €, while 37% had an income of at least 3000 € per month. Education was divided into three categories: 25.5% had “low” school education (no or secondary school certificate “Hauptschulabschluss”), 33.5% had “medium” school education (secondary school certificate “Realschulabschluss”), and 41.0% had “high” school education (university entrance qualification or university degree). A total of 80.2% of participants lived in federal states of former Western Germany (including Berlin), whereas 19.8% lived in the area of former East Germany. Also, 24.8% of the sample had children (<18 years).

### 3.2. Psychometric Properties

#### 3.2.1. Reliability

Internal consistency for the HLS-COVID-Q22 was very high (α = 0.94; ρ = 0.891). The four subscales also showed high internal consistency (subscale “access”: α = 0.814; “understand”: α = 0.858; “appraise”: α = 0.823; “apply”: α = 0.83).

#### 3.2.2. Validity

Model fit indices for a 4-factorial model representing the action areas ”access“, “understand“, ”appraise“, and “apply“ are displayed in Table 2.

All model fit indices, except SRMR, suggest an insufficient model fit for the unmodified model (Model 1). Thus, modification indices were used to revise the model. In Model 2, we allowed the residuals of items 1 and 2, 19 and 20, and 20 and 21 to correlate, which was suggested as the modifications within factors with the most significant decrease in χ² by SPSS Amos. In Model 2, the model fit indices NFI, TLI, CFI, IFI, RMSEA, and SRMR suggest a sufficient model fit, while χ²/df does not. However, if applying the more strict cut-off values recommended by Hooper and colleagues [53], only RMSEA and SRMR suggest a sufficient model fit. We refrained from further modifications to evaluate a model close to our original assumptions. Factor loadings between items and latent factors (not shown here) were 0.52–0.70 for the latent factor “access”, 0.63–0.78 for “understand”, 0.60–0.79 for “appraise”, and 0.58–0.80 for the latent factor “apply”. Correlations between the overall score and subscales are very high (r > 0.850), as well as correlations between subscales (r = 0.647 to 0.742) (Table 3).

### 3.3. Health Literacy

The health literacy mean score of the total sample was 2.99 (SD = 0.49), meaning that it was, on average, “easy” for participants to deal with information related to coronavirus. A total of 15.2% of participants were found to have “inadequate health literacy”, 34.9% had “problematic health literacy”, and 49.9% had “sufficient health literacy”. We found no significant differences in mean scores related to sex, age, education, household income, living together with children <18 years, or region of residence (Table 4). “Inadequate” and “problematic” levels were significantly more prevalent among those who felt “not so well informed/not well informed at all” about the coronavirus compared to those who reported feeling “well informed” (*p* < 0.001), or “very well informed” (*p* < 0.001). Also, “inadequate” and “problematic” health literacy levels were significantly more prevalent among those who felt “very confused” compared to those who reported feeling “somewhat confused” (*p* < 0.01), or “barely confused/not confused at all” (*p* < 0.001).

While most tasks addressed by the HLS-COVID-Q22 are easy for most respondents (mean score close to 3, Table 5), it is easiest to “find information about the coronavirus on the internet” (mean = 3.33) and “find information on the internet about protective behaviors that can help to prevent infection with the coronavirus” (mean = 3.24). On the contrary, an equal number of participants report that it is either “very difficult/difficult” or “easy/very easy” to “judge if information on coronavirus and the coronavirus epidemic in the media is reliable” (mean = 2.53), and “judge if [they] have been infected with coronavirus” (mean = 2.5), making those the most difficult tasks. Significant associations (*p* < 0.01; *p* < 0.05) for some items were observed mostly for education but partially related to age, sex, and fewer for income and having children <18 years (Appendix A).

### 3.4. Feeling Informed or Confused about the Coronavirus during the Infodemic

The vast majority (*n* = 934; 90%) feels well or very well informed about coronavirus (Figure 1). We found no significant differences related to sex, age, education, living with a child <18 years, or region. Participants with higher household income felt significantly better informed about the coronavirus (*p* < 0.05).

A total of 56.7% of the participants (*n* = 578) reported feeling confused by the variety of information on COVID-19 (Figure 2), with female participants reporting feeling more confused compared to their male counterparts (*p* < 0.01). Confusion was found to be more widespread among younger people (*p* < 0.001): 14% of people >45 years reported to feel very confused, while another 47% were somewhat confused by all the information on coronavirus, compared to 7% and 39% among people aged ≥60 years. Adults with children <18 years felt significantly more confused (*p* < 0.001). Confusion was not significantly correlated with education, income, and region. Both items were found to be negatively associated (r = −0.224; *p* < 0.001). Thus, people who feel better informed tend to be less confused about the information on the coronavirus and vice versa.

## 4. Discussion

This study represents the first population-based survey on comprehensive health literacy in relation to coronavirus information and the first to adapt the HLS-EU-Q to assess coronavirus-related health literacy. It is also the first study analyzing health literacy in the context of the COVID-19 infodemic in order to gain insights in relation to feeling informed about the coronavirus or confused by the sheer amount of coronavirus information.

### 4.1. Coronavirus-Related Health Literacy

We found that 50.1% of our sample had “problematic” (15.2%) or “inadequate” (34.9%) coronavirus-related health literacy, whereas 49.9% had “sufficient” health literacy. Mean scores show it was, on average, “easy” for participants to deal with coronavirus information. However, coronavirus-related health literacy should still be increased, as a significant share of participants score below the average health literacy level. Although an adapted questionnaire, the HLS-COVID-Q22, was used, these findings are in line with previous studies on adults’ general health literacy in Germany: The Health Literacy Survey Germany (HLS-GER) [37] found that 54.3% of the population (*n* = 2000) had “inadequate” and “problematic” health literacy, while the German Health Update (GEDA) study [45] resulted in a slightly lower estimate (44.2% of *n* = 4854). Further, our findings are comparable to what has been found in previous studies on adults’ general health literacy in Europe (47%, “inadequate” or “problematic” health literacy [28]). However, while a social gradient of health literacy has been observed in multiple studies [28,32,36,54], this was not the case in this study. The missing social gradient could be based on the sample characteristics of the original panel that might not be representative of those with lower social status. However, a missing social gradient could also indicate that ample information is available, and that the given information environment makes it easy to access, understand, appraise, and apply health information in everyday life. During the first wave of the COVID-19 outbreak, when this survey was conducted, adherence to public health interventions and government policies was very high in Germany [55,56], i.e., diverse information on coronavirus and recommendations were followed appropriately. This might also explain the missing differences in health literacy regarding sex, age, and state of residence. In addition, information and recommendations were simple and therefore easy-to-understand and easy-to-apply (e.g., hand washing, physical distance, wearing masks, avoid public gatherings, staying home). However, adherence is already declining significantly due to the various adverse effects on the economy, occupation, health, and social life [57]. This makes lower health literacy a threat to the effectiveness of public health measures to contain the virus the longer the pandemic lasts, with people less likely to follow official public health recommendations.

### 4.2. Information Tasks and Challenges

Most participants report little difficulty with accessing information about coronavirus and protective behaviors on the internet, or understanding information provided by health professionals, authorities, and family members (items # 1–3, 7–11, 15, 19–20, 22). The most significant association was found for education in relation to tasks associated with all four action areas (# 1, 2, 3, 7, 9, 14, 15, 19).

However, there are also a number of items that highlight critical aspects of dealing with coronavirus-related health information. Between ~20% and 52% of participants report that it is difficult or very difficult to access, understand, appraise, and apply coronavirus information (# 4–6, 12–14, 16–18, 21). Those “difficult” action areas are concerned primarily with medical information regarding infection (# 4, 17), help-seeking (item 5), evaluation of personal risk (# 6, 12, 14, 16), and media information (# 4, 5, 17).

For 20.9%, it is difficult to decide how to protect themselves from infection with the coronavirus based on media information. Even more citizens (32.1%) report that it is difficult to use media information to decide how to act in case of coronavirus infection. Most strikingly, 47.8% of participants report that it is difficult or very difficult to judge whether they can trust media information on the coronavirus, which is similar to findings of the HLS-GER in relation to disease-related media information [37]. Trustworthy information about the virus is vital for the population’s ability to take part in informed, healthy, and responsible behavior as well as for an effective system-wide response in line with government policies on coronavirus containment and burden reduction for the healthcare system and professionals [19,20]. Low trust in health information can cause serious problems during the pandemic, especially regarding the ongoing infodemic and the associated spread of dis- and misinformation regarding COVID-19, potentially leading to irrational behavior endangering the success of prevention measures [19]. Further, if the information on the pandemic and related preventive measures is not perceived as reliable, communication between national and local governments, including public health and research institutions on the one hand and citizens on the other hand, is impaired, casting doubt on the broad acceptance of future national and evidence-based strategies for protecting public health. Thus, there is a need for measures increasing the reliability of available and accessible coronavirus information and related actions, as well as increasing the trust of the public in such information. Another approach to enabling the general population to judge whether or not they can trust media information on the coronavirus is to promote critical health literacy [22], which can help people identify reliable sources of information and investigate which information might be misleading.

Accessing, understanding, and appraising information about risks associated with coronavirus, coronavirus infection, and adverse health behaviors pose further problems. More than 20% of participants find it difficult to understand media-based risk information about coronavirus. Deciding which behaviors can help most to lower infection risk is difficult for 22% of the population. More than 30% of adults perceive difficulties judging if they belong to a high-risk group for infection, and 33% even have difficulties finding information on risks. These results call for increased and improved risk communication, adapted to the needs of the population.

Further, it is difficult for almost 40% of the participants to find information on how to recognize a likely infection with coronavirus, and almost 52% say it is difficult to judge whether they may have been infected. These findings are worrying, as knowledge of one’s own infection status may be an important determinant of whether people adhere to distancing rules and other preventive measures [16]. Although evidence on immunity after being infected with COVID-19 is inconclusive [58], people that assume to have already been infected, e.g., based on symptoms similar to those of COVID-19, might believe to be immune and put themselves and others at risk by not adhering to preventive measures based on a false sense of safety [19]. It has been shown that even health professionals and researchers face difficulties providing accurate information on symptoms, which is why controversial information is available on the matter [8,11,59,60]. In addition, information on COVID-19 changes rapidly, which may contribute to people having difficulties judging what information is reliable [18,20]. Going forward, this could potentially jeopardize public adherence to restrictions and preventive measures, especially in the case of a second wave of infections.

Finally, participants also report having difficulties accessing information on how to find professional help in case of coronavirus infection. This can pose a serious threat in emergency situations. Therefore, efforts are needed to improve the transparency and accessibility of the healthcare system regarding diagnosis and treatment options regarding COVID-19.

### 4.3. Infodemic and Associated Influences on Information Uptake and Use

#### 4.3.1. Being Informed about Coronavirus

A total of 90% of participants feel well-informed or very well-informed about coronavirus, irrespective of sex, age, education, or living with a child <18 years. However, participants with a lower income and participants living in the federal states of former East Germany were found to feel significantly less informed. This points to the presence of an economic as well as a regional gradient. While our data does not provide further clues on reasons for these differences, a number of reasons are plausible. Regarding economic differences, it is possible that the information on coronavirus available is sufficient for the majority, but that low-income participants have specific information needs that are not fully satisfied by the information circulating in major news channels. For instance, specific questions might arise more frequently among low-income participants during the pandemic concerning topics such as unemployment, social benefits, and social security, or youth services. The regional differences, however, may result from differences in patterns and frequencies of internet use in eastern and western Germany. While the latest Eurostat data shows that 9 out of every 10 households have internet access [61], there are differences in internet use between households in the east and west of Germany with people living in eastern regions using the internet significantly less than their counterparts [62]. Another explanation could be that the outbreak of COVID-19 started in the west of Germany [63], which is why at the time of this survey, people living in the east of Germany may have felt less concerned about coronavirus and less engaged with seeking information.

However, the overall level of feeling informed is high, which might, at least partly, be attributed to government-led public health and media campaigns early on in the pandemic using different channels including social media. This finding may also reflect how well German public health authorities and public broadcasting agencies have reacted to the pandemic and associated challenges for society and individuals.

#### 4.3.2. Being Confused about Coronavirus

Despite the findings, more than half of the German population (56%) feels somewhat confused or very confused about the amount of coronavirus information. Although it was expected that confusion and feeling informed were diametrically opposed indicators, this was not always the case. Even though a negative correlation was found between those variables, it was only small in size (r = −0.224). This means that there are a number of participants that feel well-informed and, at the same time, confused by all the information. Hence, it is not only necessary to provide citizens with coronavirus information they need so that they feel well-informed, but also to help them make sense of and evaluate additional and potentially conflicting information to avoid confusion.

We found that women reported feeling confused by the amount of coronavirus information more often than men. While the virus and virus information are not gender-specific, it may be related to the unequal distribution of care responsibilities, which is still prevalent in Germany [64]. As women in Germany are more often engaged in the upbringing of children, providing childcare and care of elderly family members [64,65], and more likely than men to seek online health information [66], it is possible that they are confronted more often with confusing coronavirus information.

Further, we found that younger people were more confused than older people by coronavirus information. This can partly be explained by a more prevalent media use among younger people and thus the higher exposure to a broad range of potentially conflicting information, but there are probably other reasons that are unaccounted for in our data, such as frequently using social media websites (e.g., Twitter, Facebook, and Instagram), reading more about coronavirus, having fear of COVID-19, being stressed by the pandemic, and feeling uncertain regarding their future and the many changes that arise from the impact of COVID-19 on their lives.

#### 4.3.3. Health Literacy and Infodemic

Regarding the relationship between feeling informed or confused and health literacy, we found that people with “inadequate” and “problematic” health literacy were significantly less likely to feel well-informed and more likely to feel confused by the amount of coronavirus information. It has to be noted that assumptions about cause and effect are difficult here, as “feeling informed” could be regarded as equivalent to two main dimensions of health literacy as measured by the HLS-COVID-Q22, namely the perceived ease in accessing and understanding health information. The same could be assumed for “feeling confused” and another core dimension of health literacy, namely the perceived ease or difficulty in appraising health information.

It has been highlighted that health education and teaching health literacy are critical prevention and health promotion measures for mitigating the adverse effects of the COVID-19 infodemic [10,13]. The benefits of fostering health literacy, the acquisition of competencies, and the confidence to handle the sheer amount of information in an infodemic are not restricted to infectious diseases. Moreover, they will also be helpful in different public health areas, such as increasing critical thinking, promoting healthy behavior and managing risk behavior, informed decision making, empowerment, and health outcomes, as well as improving health care interaction, health communication, and adherence.

In order to facilitate health information seeking and understanding, information suppliers and providers, such as official public health organizations and health authorities, have a critical role in providing the public with high-quality health information [59]. When designing and providing coronavirus and COVID-19-related information, providers must ensure that information is based on health literacy principles such as being easy-to-access, easy-to-understand, easy-to-use, culturally appropriate, and relevant to various populations [20]. Accessing valid and reliable health information on the internet is among the greatest challenges for internet users [60,67]. Social media providers, who are important actors in the infodemic [4,11,68,69,70,71,72], must be urged to act responsibly, support the provision of reliable health information, and inhibit the spread of dis- and misinformation on their websites [13]. In this regard, government support is needed to implement policies that hold social media and tech giants accountable, e.g., by fact-checking and flagging false information.

The perpetual stream of information overload that is driven by actors across society amplifies the COVID-19 infodemic. The infodemic must be acknowledged as a meta-risk in its own right, which interferes with people’s health literacy and their health outcomes. Limited health literacy in individuals, populations, and systems can cause adverse effects for parts and even the whole of society. When people lack the competencies to critically appraise health information, suppliers cannot ensure the means to protect valid and meaningful information against the many sources that spread invalid information. This can cause panic, destabilize the effectiveness of information distribution and public health interventions, and even threaten social cohesion and the political landscape. As Eysenbach highlighted in the context of infodemiology studies, “an epidemic of fear may exhibit similar characteristics as a true epidemic” [73], which is why investment in population and system-level health literacy must be considered absolutely critical for public health responses against toxic health threats, such as coronavirus and COVID-19. This is supported by a recent study on health literacy and fear of COVID-19 in medical students, showing that higher levels of health literacy may protect from fear during the pandemic [38].

### 4.4. Feasibility of the Questionnaire

The HLS-COVID-Q22 is a novel tool in the HLS-family and the first to measure coronavirus-related health literacy. Indicators for internal consistency demonstrate satisfactory reliability (α = 0.940 and ρ = 0.891), suggesting the success of the adaptation process and the feasibility of the instrument. Internal consistency estimates found in this study are in line with what has been found in previous studies using the HLS-EU-Q16 in German language (α = 0.96) [45], which served as the original blueprint for developing the HLS-COVID-Q22. Internal consistency for the four subscales was also satisfactory, which means that they can be used to assess subjective coronavirus-related health literacy related to the areas of accessing, understanding, appraising, and applying coronavirus-related health information.

Construct validity for the four subscales suggested that the model fit was only moderate. However, the model fit indices clearly did not indicate an excellent model fit, neither based on the moderate cut-off values [49] that were applied nor based on strict cut-off values as suggested by Hooper and colleagues [53]. An acceptable model fit was only achieved in some indicators after a slight modification. In this modification step, we allowed the residuals of items 1 and 2, 19 and 20, and 20 and 21 to correlate, expressing a common source of variance. These modifications were acceptable from a theoretical and content-related standpoint, as those item pairs are concerned with finding information on the internet (items 1 & 2), following or using information received from the doctor (items 19 & 20), and using information on how to handle an infection with coronavirus (items 20 & 21). After allowing the correlation between the item’s residuals using moderate model parameters [49], the model fit indices NFI, TLI, IFI, CFI, RMSEA, and SRMR of Model 2 suggested a sufficient model fit, whereas χ^2^/df did not. To reach an excellent model fit based on both the strict [53] or the more moderate [49] values employed here, further modifications or even a revision of the underlying model might be necessary. However, we chose not to apply further modifications, such as correlations of residuals across factors (as suggested by SPSS Amos), as this would have implied further, potentially atheoretical deviation from the model which suggests 4 distinct factors. However, the high correlations (r = 0.65–0.90) found between the subscales (and thus, factors) conflict with the assumption of four independent factors and point to a second-order common factor, i.e., content-wise similarities between the highly correlated factors. This might contribute to less-than-optimal model fit and warrant further investigation of the underlying factor structure.

Earlier studies with the HLS-EU-Q have reported similar problems regarding model fit for the four-factorial model, assuming a rather unidimensional model [74]. Unfortunately, most studies using the HLS-EU-Q do not provide data on construct validity. In a Norwegian study, the model fit was reported for the HLS-EU-Q16 for a one- and three-dimensional model with values for TLI = 0.911 and 0.939, CFI = 0.923 and 0.949, SRMR = 0.080 and 0.070, and RMSEA = 0.118 and 0.103, respectively [74]. In a study conducted in six Asian countries using the HLS-EU-Q47 [36], the model fit indices were reported for a three-factorial model representing the areas of health promotion, disease prevention, and health care. In each domain, the four-factor model then included accessing, understanding, appraising, and applying health information. For all six countries, RMSEA values ranged from 0.05–0.10, CFI ranged from 0.90–0.97, IFI ranged from 0.90–0.97, NFI ranged from 0.87–0.96, χ^2^/df ranged from 2.55–21.85. In a Japanese study using the HLS-EU-Q47, CFI and RMSEA were reported for the three domains with values for CFI 0.937 and RMSEA 0.075 (health care), 0.943 and 0.079 (disease prevention), and 0.934 and 0.078 (health promotion) [75]. The model fit indices of the HLS-COVID-22 are comparable to earlier studies. This calls for a more detailed approach to operationalising the four action areas, better reflecting the actions taken to achieve them, i.e., deconstructing the steps necessary to access information.

This study was conducted during the first weeks of the pandemic in Germany while people were already overburdened and stressed with the many changes they had to cope with and apply in everyday life and at the workplace. Therefore, we decided to keep the questionnaire rather short and did not include further scales and variables, such as COVID-19-related health knowledge and behavioral aspects. Based on the HLS-EU-Q, the HLS-COVID-Q22 is a self-report measure and does not assess the performance-based health literacy capabilities of individuals [76]. For a better understanding of the associations and possible relationships between health literacy on the one hand and knowledge as well as health behavior on the other, we plan to include additional questions to the follow-up surveys (waves 2 and 3 of this measurement) to allow comparisons between subjective health literacy and objective performance-based skills and abilities. These waves will be implemented in 2020 as a three-country survey in Germany, Austria, and Switzerland in order to analyze change over time (German Federal Ministry of Health, grant number: ZMVI1-2520COR009).

## 5. Limitations

This study was conducted during the first wave of the COVID-19 outbreak in Germany when information on the topic was easy to understand and omnipresent in the media and internet, supported by government campaigns and campaigns by public health agencies. This may have contributed to the overall good results, i.e., the high satisfaction with information and the perceived ease regarding most information tasks. These results may not be transferred to other countries that use different public information and communication strategies. Future studies should also consider measuring behavioral accounts and assessing the health literacy of different populations, e.g., children, students, migrants, patients, and people with chronic conditions. Further, coronavirus-related health literacy was assessed by employing a self-report measure, which generally is a reliable method to measure subjective indicators. However, participants’ actual ability to access, understand, appraise, and apply coronavirus-related health information was not tested, and it is up to future research to determine to what extent the indicators “perceived ease” and “ability” correspond. Also, future research may need to include additional scales to widen the scope of analysis. Lastly, the study sample is representative of German internet users, which excludes non-users who may feel different about the infodemic.

## 6. Conclusions

The HLS-COVID-Q22 has proven to be a feasible and reliable tool to assess adults’ coronavirus-related health literacy. During the first wave of the COVID-19 pandemic and the associated infodemic, more than half of the German population was found to have “inadequate” or “problematic” levels of health literacy. The judgment of media-based information on COVID-19, especially, seems to be difficult. There is a need for promoting critical health literacy needed to navigate the infodemic, identify disinformation and misinformation, and make decisions based on reliable and trustworthy information. Political attention must be paid to policies supporting the enhancement of population health literacy and sustaining easy-to-access and easy-to-use information. More research on health literacy will facilitate a better understanding of population information needs, i.e., with further surveys, monitoring and surveillance systems, and longitudinal studies for informing public health and policymaking. Health literacy should also be considered in the COVID-19 public health response framework and public health should be prompted to further research health literacy with policies to support such endeavours. As COVID-19 has put an unprecedented burden on the healthcare system and workforce, addressing health literacy in such a response framework will be of great value for healthcare, health promotion, and prevention. Information supply as a public health intervention should be considered in future emergency and non-emergency response frameworks. Additionally, linking research on health literacy with evidence from information literacy, media literacy, risk literacy and infodemiology may help to widen the understanding how people seek for and use information in everyday life.

## Figures and Tables

**Figure 1 ijerph-17-05503-f001:**
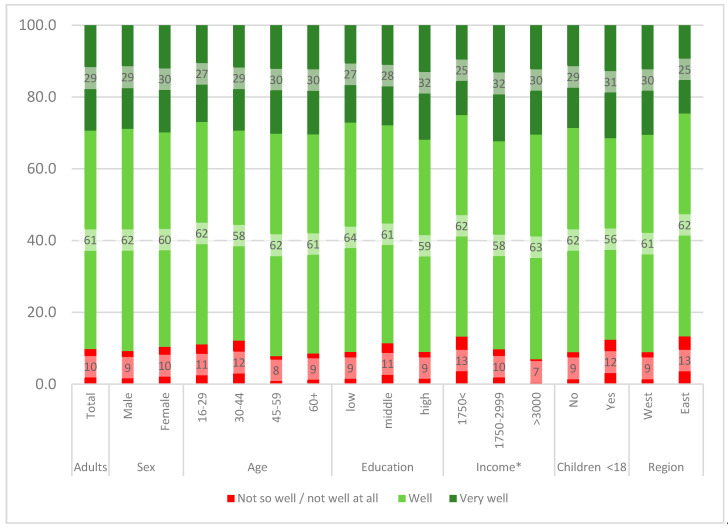
How well are you informed about coronavirus? * *p* < 0.05; Spearman correlation, *n* = 1037, see Appendix A.

**Figure 2 ijerph-17-05503-f002:**
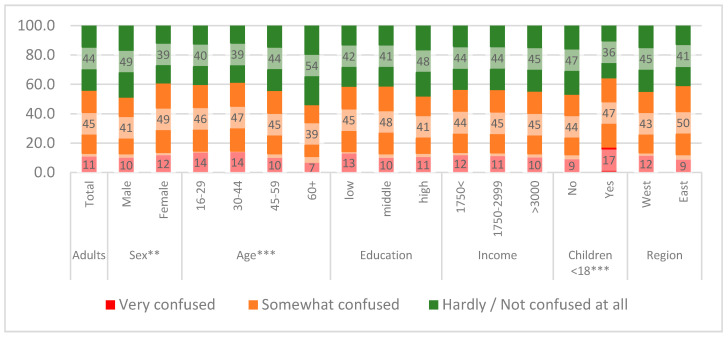
Do you feel confused about coronavirus information? ** *p* <.01; *** *p* < 0.001; Spearman correlation, *n* = 1037, see Appendix A.

**Table 1 ijerph-17-05503-t001:** Health literacy scores and cut-off values.

Tools	# of Items	Score	Answers Dichotomized *	Scores Transformed †	Cut-Off Values (Relative)	Cut-Off Values (Absolute)
HLS-EU-Q47 [28,42]	47	Mean	No	Yes	a ≤ 50% (½) 50% < b < 66% (~⅔) 66.6% ≤ c < 84% (~⅚) 84 ≤ d	0 < a ≤ 25 25 < b < 33 33 ≤ c < 42 42 ≤ d
HLS-EU-Q16 [44]	16	Sum	Yes	No	a ≤ 50% 50% < b < 81.3% 81.3% ≤ c	0 < a < 9 9 ≤ b < 13 13 ≤ c
HLS-COVID-Q22 [46]	22	Mean	No	No	≤ 50% 50% < b < 66% 66% ≤ c	a ≤ 2.5 2.5 < b < 3 3 ≤ c

Notes. * = In their study, Röthlin et al. [44] dichotomized the answers to “very difficult/rather difficult” vs. “rather easy/very easy”; † = Transformed with the formula y = (mean − 1) × (50 ÷ 3); a = “inadequate health literacy”; b = “problematic health literacy”; c = “sufficient health literacy”; d = “excellent health literacy”.

**Table 2 ijerph-17-05503-t002:** Model fit indices for a 4-factorial model representing the HLS-EU model structure [23,42].

Model	χ^2^/df	NFI	IFI	TLI	CFI	RMSEA [95% CI]	SRMR
Model 1	7.664	0.872	0.887	0.871	0.887	0.080 [0.077–0.084]	0.048
Model 2	5.996	0.902	0.917	0.904	0.916	0.069 [0.066–0.073]	0.043

**Table 3 ijerph-17-05503-t003:** Correlations between the overall health literacy score and the subscales.

Questionnaire and Subscales	Subscale “Access”	Subscale “Understand”	Subscale “Appraise”	Subscale “Apply”
HLS-COVID-Q22	0.880 **	0.897 **	0.876 **	0.874 **
Subscale “access”	1	0.742 **	0.690 **	0.647 **
Subscale “understand”	-	1	0.675 **	0.736 **
Subscale “appraise”	-	-	1	0.729 **

Notes. ** = *p* < 0.01 (two-tailed). *n* = 1037 for all correlations (Pearson coefficient).

**Table 4 ijerph-17-05503-t004:** Health literacy levels and mean scores by participant characteristics.

Study Population	Health Literacy Levels [%]			Health Literacy Scores	
	“Inadequate”	“Problematic”	“Sufficient”	Mean (SD)	*p*
**Total** (***n* = 1037**)	15.2	34.9	49.9	2.99 (0.49)	
**Sex**					
Female (*n* = 505; 48.7%)	15.2	34.5	50.3	2.99 (0.49)	ref.
Male (*n* = 532; 51.3%)	15.2	35.2	49.6	2.99 (0.49)	0.904
**Age**					
16–29 (*n* = 226; 21.8%)	14.8	39.0	46.2	2.96 (0.48)	0.326
30–44 (*n* = 267; 25.7%)	15.9	31.4	52.7	2.99 (0.50)	ref.
45–59 (*n* = 319; 30.8%)	15.2	32.6	52.2	3.02 (0.50)	0.877
60+ (*n* = 225; 21.7%)	14.8	38.0	47.2	2.98 (0.48)	0.614
**Education**					
Low (*n* = 264; 25.5%)	18.0	35.8	46.1	2.94 (0.52)	0.597
Medium (*n* = 347; 33.5%)	15.3	35.6	49.2	2.98 (0.49)	ref.
High (*n* = 425; 41%)	13.4	33.7	52.9	3.03 (0.48)	0.161
**Household income**					
Below 1750 € (*n* = 327; 31.5%)	18.1	34.4	47.6	2.96 (0.52)	0.496
1750–2999 € (*n* = 327; 31.5%)	13.9	34.8	51.3	3.00 (0.48)	ref.
3000 € + (*n* = 383; 37%)	13.9	35.4	50.8	3.01 (0.48)	0.409
**Do you have children younger than 18?**					
Yes (*n* = 257; 24.8%)	18.6	30.4	51.0	2.96 (0.54)	ref.
No (*n* = 780; 75.2%)	14.1	36.3	49.6	3.00 (0.48)	0.633
**Region**					
Former Western Germany (*n* = 832; 80.3%)	15.6	33.6	50.8	2.99 (0.49)	ref.
Former East Germany (*n* = 205; 19.7%)	13.7	39.9	46.4	2.97 (0.49)	0.322
**How well are you informed about the coronavirus?**					
Very well (*n* = 304; 29.3%)	4.7	28.3	67.0	3.23 (0.50)	<0.001
Well (*n* = 631; 60.9%)	13.6	39.7	46.7	2.95 (0.41)	ref.
Not so well/not well at all (*n* = 102; 9.8%)	56.4	24.5	19.1	2.51 (0.51)	<0.001
**Do you feel confused about COVID-19 information?**					
Very confused (*n* = 115; 11.1%)	40.2	28.2	31.6	2.72 (0.57)	<0.01
Somewhat confused (*n* = 463; 44.6%)	17.0	42.4	40.6	2.90 (0.43)	ref.
Barely confused/not confused at all (*n* = 459; 44.3%)	7.1	28.9	64.0	3.15 (0.47)	<0.001

Notes. ref. = reference group; *p* = asymptotic significance (2-tailed). To transfer mean scores to frequently reported metric ranging from 0–50, this formula can be used: y = (mean − 1) × (50 ÷ 3). The inverse formula, y = (3 × mean ÷ 50) + 1, transforms scores on the transformed metric to mean scores on the original response format (1–4).

**Table 5 ijerph-17-05503-t005:** Frequencies and mean scores for HLS-Covid-Q22 items.

#	How Easy or Difficult is It for You to…	Related HLS-EU-Q Item ^a^	Very Difficult	Difficult	Easy	Very Easy	Mean (SD)
1	Find information about the coronavirus on the internet?	1; 2	1.2	6.4	49.7	42.7	3.34 (0.65)
2	Find information on the internet about protective behaviors that can help to prevent infection with the coronavirus?	13; 33	1.1	9.9	52.4	36.6	3.25 (0.67)
3	Find information in newspapers, magazines and on TV about behaviors that can help to prevent infection with the coronavirus?	13; 33	2.3	16.3	54.5	26.8	3.06 (0.72)
4	Find information on how to recognize if I have likely become infected with the coronavirus?	8; 18	6.2	27.6	46.7	19.6	2.80 (0.82)
5	Find information on how to find professional help in case of coronavirus infection?	2; 4	6.0	27.7	46.1	20.2	2.81 (0.83)
6	Find information on how I much I am at risk for being infected with the coronavirus?	2; 4	4.1	28.9	46.4	20.6	2.84 (0.79)
7	Understand your doctor’s, pharmacist’s or nurse’s instructions on protective measures against coronavirus infection?	4; 8	1.0	9.7	57.7	31.5	3.20 (0.65)
8	Understand recommendations of authorities regarding protective measures against coronavirus infection?	10; 23	2.6	13.0	53.7	30.7	3.13 (0.73)
9	Understand advice from family members or friends regarding protective measures against coronavirus infection?	14; 37	2.2	12.5	58.3	27.0	3.10 (0.69)
10	Understand information in the media on how to protect myself against coronavirus infection?	15; 39	1.3	11.2	58.3	29.2	3.15 (0.66)
11	Understand risks of the coronavirus that I find on the internet?	9; 21	1.8	17.1	55.2	25.9	3.05 (0.71)
12	Understand risks of the coronavirus that I find in newspapers, magazines or on TV?	3; 5	2.3	17.8	56.0	23.9	3.01 (0.71)
13	Judge if information on the coronavirus and the coronavirus epidemic in the media is reliable?	11; 28	11.3	36.5	39.0	13.1	2.54 (0.86)
14	Judge which behaviors are associated with a higher risk of coronavirus infection?	16; 43	3.6	18.4	53.6	24.4	2.99 (0.76)
15	Judge what protective measures you can apply to prevent a coronavirus infection?	5; 11	2.6	15.5	54.4	27.4	3.07 (0.73)
16	Judge how much I am at risk for a coronavirus infection?	11; 43	3.8	27.4	45.8	23.0	2.88 (0.80)
17	Judge if I have been infected with coronavirus?	11; 43	11.3	40.5	34.8	13.4	2.50 (0.86)
18	Decide how you can protect yourself from coronavirus infection based on information in the media?	12; 31	3.6	17.3	57.5	21.6	2.97 (0.73)
19	Follow instructions from your doctor or pharmacist regarding how to handle the coronavirus situation?	7; 16	2.1	12.0	58.1	27.8	3.11 (0.69)
20	Use information the doctor gives you to decide how to handle an infection with the coronavirus?	6; 13	2.3	17.0	59.4	21.3	3.00 (0.69)
21	Use media information to decide how to handle an infection with the coronavirus?	6; 13	4.2	27.9	50.3	17.6	2.81 (0.77)
22	to behave in a way to avoid infecting others?	12; 31	2.5	11.9	52.1	33.5	3.17 (0.73)

Notes. Items were used in German but have been translated for the purpose of reporting; health literacy action areas: access (#1–6), understand (#7–12), appraise (#13–17), and apply health-related information (#18–22); ^a^ = the first number indicates based on what item in the HLS-EU-Q16 the HLS-Covid-Q22 item was adapted, the second number relates to the item number in the HLS-EU-Q47.

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
