# Peer review of "Coronavirus-Related Health Literacy: A Cross-Sectional Study in Adults during the COVID-19 Infodemic in Germany"

_ijerph, 2020, doi:10.3390/ijerph17155503_

Round 1

Reviewer 1 Report

This study looks very important and carefully structured and efficiently describes the health literacy about covid-19 in Germany.

However, I felt one, but not small, problem in the logic of this study.

Infodemic may make people confused in finding and understanding the right information.  In addition, it can lead people to false perception that they could find and access the right information.

The questionnaire used in this study assesses the perception of difficulty or easiness to find the information as health literacy, but doesn't assess whether the information they found is correct or not.  About covid-19, even specialists sometimes state wrong information.  Therefore, to validate the questionnaire of health literacy, the correctness of the knowledge and/or understanding of the fact related to covid-19 should also be evaluated.

At least, the authors should give discussion about why such objective evaluation of the respondent's knowledge and/or understanding of the covid-19 related facts was not done.

Author Response

Dear Reviewer,

Thank you so much for your precise and positive review of our research study on coronavirus related health literacy during the infodemic. We have applied your suggestion accordingly and where we saw fit. Please see below our point-by-point response to your queries.

Comments and Suggestions for Authors

This study looks very important and carefully structured and efficiently describes the health literacy about covid-19 in Germany.

Thank you

However, I felt one, but not small, problem in the logic of this study.

Infodemic may make people confused in finding and understanding the right information.  In addition, it can lead people to false perception that they could find and access the right information.

The questionnaire used in this study assesses the perception of difficulty or easiness to find the information as health literacy, but doesn't assess whether the information they found is correct or not.  About covid-19, even specialists sometimes state wrong information.  Therefore, to validate the questionnaire of health literacy, the correctness of the knowledge and/or understanding of the fact related to covid-19 should also be evaluated.

At least, the authors should give discussion about why such objective evaluation of the respondent's knowledge and/or understanding of the covid-19 related facts was not done.

We have included a new paragraph to the discussion section where we talk about the questionnaire (see p 14, lines 491-497). We do briefly explain the reasons to keep the questionnaire short (early-on in the pandemic with people already overburdened and stressed out be the changes in their live). In the planned second and third waves we are going to include such questions. In addition, we also added a sentence to the limitation section.

We hope to have met your expectations regarding the revision. Thank you so much.

Best wishes,

The authors

Reviewer 2 Report

Page2, paragraph 2, lines 62-70--please revise: sentence structure and clarity

Page 2, paragraph 2.1, lines 83-91: please make it clear to the reader why this study was conducted since there was a great deal of information already; that did not become clear to this reader/reviewer until the discussion section in which the authors described the need for an updated assessment. 

Page 3, paragraphs 1-2, lines 99-113: you stated that the sample is representative of the population. It would be helpful to the reader/reviewer to have reference to the table of data in which you demonstrate representation

Page 3, paragraph 2.2.2 and 2.2.3: Why are there no fact-based questions on the questionnaire that would help to explain (or not) the respondent's sense of clarity or confusion, etc.?

Data analyses and results are thoroughly presented and discussed. 

In tables of results and in text, there is some variation in the use of comma versus period as separator. 

The presentation of an infodemic is well founded and is applicable across many nations. The need for correct information, fact-checking, and periodic updating by experts as the pandemic progresses in populations is well founded and is indeed a critical need 

Author Response

Dear Reviewer,

Thank you so much for your precise review. We have applied your suggestions accordingly and where we saw fit. Please see below our point-by-point response.

  • Page2, paragraph 2, lines 62-70--please revise: sentence structure and clarity

The respective paragraph has been revised to increase the flow of reading.

  • Page 2, paragraph 2.1, lines 83-91: please make it clear to the reader why this study was conducted since there was a great deal of information already; that did not become clear to this reader/reviewer until the discussion section in which the authors described the need for an updated assessment. 

The respective sentence (p 2, lines 89-93) has been revised in order to more clearly highlight the relevance. In addition, the whole two paragraphs on page 2, lines 64-94 draw on why this study is important, which we believe will support reader comprehension of the study relevance. In order to remain a clear structure and coherent line of argumentation, we`d appreciate to not make any further changes.

  • Page 3, paragraphs 1-2, lines 99-113: you stated that the sample is representative of the population. It would be helpful to the reader/reviewer to have reference to the table of data in which you demonstrate representation

This is a very good point and we have included the reference to table 4 which is where we describe the study sample. Journal requirements clearly state to include the table when first referring to it. However, this table 4 belongs to where it currently is. If the journal has no objection, we are happy with this revision.

  • Page 3, paragraph 2.2.2 and 2.2.3: Why are there no fact-based questions on the questionnaire that would help to explain (or not) the respondent's sense of clarity or confusion, etc.?

We have included a new paragraph to the discussion section where we talk about the questionnaire (see p 14, lines 491-497), where we briefly explained the reasons to keep the questionnaire short (early on in the pandemic with people already overburdened and stressed out be the changes in their live). In the planned second and third waves we are going to include such questions. In addition, we also added a sentence to the limitation section.

  • Data analyses and results are thoroughly presented and discussed. 

Thank you.

  • In tables of results and in text, there is some variation in the use of comma versus period as separator. 

This has been revised.

  • The presentation of an infodemic is well founded and is applicable across many nations. The need for correct information, fact-checking, and periodic updating by experts as the pandemic progresses in populations is well founded and is indeed a critical need 

Thank you.

Best wishes,

The authors

Round 2

Reviewer 1 Report

Limitation of the study and its reason are clearly described now.